# Hazop Analysis of a Bioprocess for Polyhydroxyalkanoate (PHA) Production from Organic Waste: Part B

Roberto Lauri [1,*], Emma Incocciati [2], Biancamaria Pietrangeli [3], Lionel Nguemna Tayou [4], Francesco Valentino [5], Marco Gottardo [5] and Mauro Majone [4]

[1] Inail, Department of Technological Innovations and Safety of Plants, Products and Human Settlements, Via del Torraccio di Torrenova 7, 00133 Rome, Italy
[2] Inail, Advisory Department for Risks Assessment and Prevention, Via Roberto Ferruzzi 38/40, 00143 Rome, Italy
[3] Inail, Department of Technological Innovations and Safety of Plants, Products and Human Settlements, Via Roberto Ferruzzi 38/40, 00143 Rome, Italy
[4] Department of Chemistry, "La Sapienza" University of Rome, P.le Aldo Moro 5, 00185 Rome, Italy
[5] Department of Environmental Sciences, Informatics and Statistics, Cà Foscari University of Venice, Via Torino 155, 30170 Mestre-Venice, Italy
* Correspondence: r.lauri@inail.it; Tel.: +39-062-094-3212

**Abstract:** The production of polyhydroxyalkanoates (PHAs) from industrial waste streams has attracted the attention of researchers and process industries because they could replace traditional plastics. The integrated treatment of civil wastewater along with organic solid wastes in a novel "urban biorefinery" is a very important option to implement a synergic treatment of all relevant bio-waste streams. The development of new biotech processes should consider the occupational health and safety issues from the initial design stages. Among the process hazards analysis techniques, HAZard and OPerability (HAZOP) methodology is widely used for studying both the processes hazards and their operability problems, by exploring the effects of any deviations from design conditions. In the present study, a modified version of HAZOP methodology has been applied to a three-step process developed at pilot scale in the Treviso municipal wastewater treatment plant in order to produce PHA. This paper (part B) shows the results of HAZOP analysis applied to the second process step aimed at culturing the activated sludge under periodic feeding conditions into a sequencing batch reactor (SBR). The analysis applied to the process conditions, corresponding to the maximum PHA content in the biomass, has led to the identification of technical solutions to mitigate the main occupational risks.

**Keywords:** polyhydroxyalkanoates; HAZOP analysis; occupational health and safety; biological risk assessment; operability; P&ID; deviation; cause; consequence

## 1. Introduction

The issue of waste generation in the form of wastewater and solid wastes has caused an urgent, yet persisting, global issue, which needs the development of sustainable treatment and resource recovery technologies.

The production of polyhydroxyalkanoates (PHAs) from industrial waste streams could replace traditional plastics. The increasing availability of raw renewable materials, the rising demand and use of biodegradable polymers for bio-medical, packaging and food applications, along with favorable green procurement policies are expected to benefit PHA market growth.

Furthermore, because of the environmental awareness and the subsequent legislations on prohibition or reduction of plastic materials, such as the European Parliament Directive on single use plastics (EUPPD, 2019), the production of bioplastic and PHAs, in particular, is rapidly increasing. The global PHA market is expected to reach USD 167 million by

2027, characterized by an annual growth rate (CAGR) of 15.3% from USD 81 million in 2022 [1]. PHAs are biopolymers with high degradability and a variety of applications in the manufacturing sector.

Nowadays, the production of PHA at industrial scale is carried out using bacterial strains with high costs associated with the use of refined sugar substrates and the adoption of axenic operation conditions. However, an increasing trend for deployment of mixed microbial communities (MMC) is observed in the last decade. The possibility of consuming a wider range of carbon sources, such as agricultural and industrial wastes, fewer process control parameters and lower operational and maintenance costs due to lack of sterilization put MMC in a superior position in PHA production processes compared to pure cultures. A number of accumulating microorganisms, such as Azoarcus spp., Thauera spp., Paracoccus spp., Zoogloea and Plasticicunulans, has been identified as prominent players in PHA production [2–4]. In mixed culture operations, a selective environmental pressure can result in obtaining a biomass with high PHA storage potential.

Notwithstanding the undoubted advantages, which derive from the commercialization of these polymers, there are no specific techniques for hazard identification in PHA production bioprocesses, especially addressed to occupational health and safety. The conventional hazard identification techniques may often overlook the specific issues posed by biological reactions [5].

A specific methodology for hazard identification of major accidents, addressed to biohazards and conventional chemical hazards in the framework of process safety, has been applied by Casson Moreno and Cozzani to the anaerobic digestion of animal manure for biogas production [6].

Among the process hazard analysis techniques, hazard and operability (HAZOP) methodology is diffusely used for studying both the process hazards and its operability problems, by exploring the effects of any deviations from design conditions.

The mentioned methodology is based on a specific layered approach, which allows us to identify the industrial bioprocess hazards, and it consists of two levels: a checklist and a HAZOP analysis, modified to consider both engineering and biotechnological aspects, and their interactions (BioHazOp). The tool goal is to foster the integration of biotechnological aspects and conventional chemical engineering process hazards, and the identification of their cause–consequence relations.

In the present study, a modified version of HAZOP methodology has been applied to a three-step process aimed at producing PHA as final high-value product. The process has been developed at pilot scale in the Treviso municipal wastewater treatment plant (WWTP) [7]. In particular, the PHA production by MMC, using pretreated organic fraction of municipal solid waste (OFMSW) and sewage sludge (SS), is based on the feast−famine approach in the traditional three-steps process scheme. The analysis has been applied to process conditions corresponding to the maximum PHA content in the biomass [8].

The paper is focused on the results of BioHazOp analysis applied to the second step (SBR) of PHA production process. As a result of the HAZOP analysis application, an in-depth assessment of occupational biological risk has been carried out and an approach for its control has been proposed.

A piping and instrumentation diagram (P&ID), which shows the interconnection of equipment and the instrumentation required for controlling the process, has been drawn for the SBR. The diagram is the basis for the development of system control schemes, allowing safety and operational investigations.

## 2. Materials and Methods

### 2.1. HAZOP Analysis

The HAZOP safety analysis technique is applied worldwide and recognized by legislation since it has demonstrated its effectiveness in identifying environmental, safety and health hazards. A HAZOP study is a highly disciplined procedure aimed at identifying how a process may deviate from its design intention. This method is able to distinguish between

hazard (any operation, that could cause a catastrophic release or that could result in harm to workers) and operability (any operation within the project that could cause a plant shutdown with possible impact on safety or profits). It is defined as the application of a formal, systematic and critical examination of the process and the engineering intentions of new or existing facilities to assess the outcomes of possible operating failures of individual equipment pieces and the consequential effects on the facility.

Its success is based on the strength in process flow diagrams (PFDs) and Piping and Instrumentation Diagrams (P&IDs), breaking the design into manageable sections with definite boundaries called nodes, so ensuring the analysis of each piece of process equipment. A small multi-disciplinary team undertakes the analysis, whose members should have sufficient experience and knowledge to answer most questions.

The method relies on the use of guidewords (such as no, more, less) combined with process parameters (e.g., temperature, flow, pressure), which aims at revealing deviations (such as less flow, higher temperature) from the process intention or normal operation. This procedure is applied to a particular node as part of the system characterized by nominal intention of the operative parameters. After the deviations have been determined, the expert team explores their feasible causes and their possible consequences. For every pair of cause-consequences, safeguards, which could prevent, detect, control or mitigate the hazardous situation, must be identified. Finally, if the safeguards are insufficient to solve the problem, additional prevention and protection measures and recommendations must be considered [9].

The HAZOP methodology is applicable to all types of installations at any stage of their life, particularly for new installations, where operational experience is lacking.

The study objectives should be made as clear as possible; in general, the goal is to identify risks and operational failures.

Among others, the reasons for HAZOP analysis may be:

- To verify the project safety;
- To check operating and safety procedures;
- To increase the safety of an existing system;
- To verify that safety equipment is working in the best possible way.

A HAZOP analysis can be split into five stages:

(1) Definition of the purpose and objectives of the study;
(2) Team selection;
(3) Study preparation;
(4) Carrying out the analysis;
(5) Recording the results.

This study result may lead to a project revision and/or check through the following actions:

- Operating procedures development;
- Verification of design values, process parameters and possible modifications;
- Request for additional alarms;
- Request for unforeseen alarms or blocks;
- Useful information for assessing and managing the risk associated with the accidental identified events.

The team members should be expert in the most relevant areas for plant operation.

*2.2. HAZOP Methodology*

The basic requirements of the HAZOP and BioHazOp analysis are:

- The examination must be systematic;
- The analysis must be carried out with a degree of formality (e.g., forms), so that the reasons for each decision made during the analysis can be clearly identified by different people at different times;
- All working on the implementation of the project must be involved in the analysis.

In biotech processes as well as in traditional chemical ones, the hazard identification plays a critical role, since all unidentified hazards could generate uncontrolled risks.

Based on the plant documentation (diagrams, cause–effect matrices, data sheets), all lines/items are analysed through the iterative process shown in Figure 1.

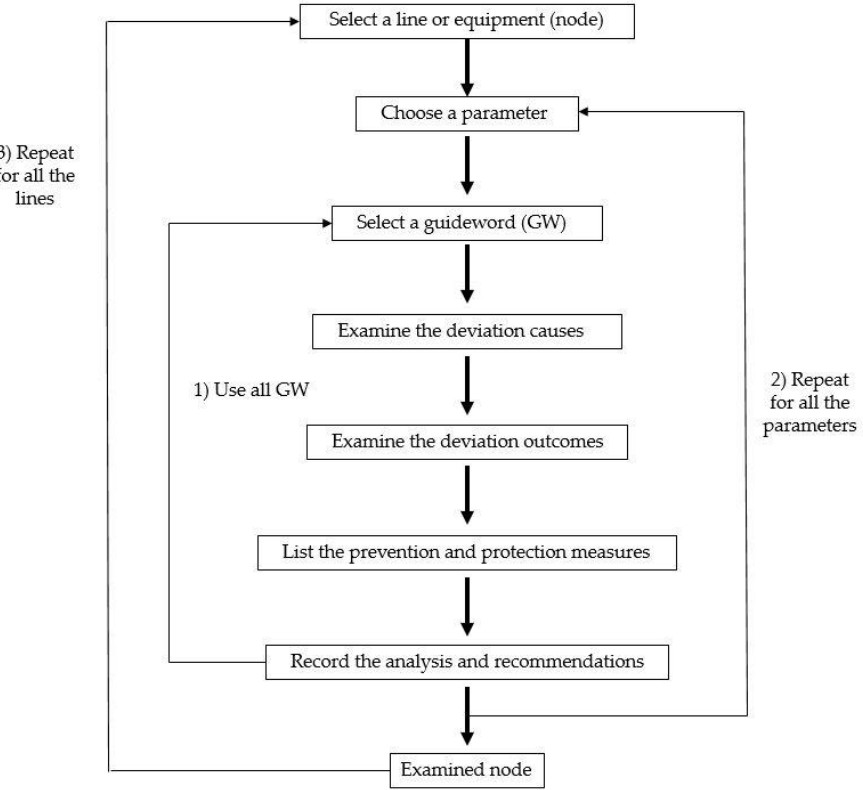

**Figure 1.** HAZOP flowsheet.

The layered approach to hazard identification is aimed at analyzing simultaneously both conventional chemical processes hazards and specific bioprocesses hazards by an integrated procedure [10]. In the case study, two specific tools were used:

(1) A checklist;
(2) A tailored HAZOP method (BioHazOp).

It should be pointed out that, as in conventional risk assessment, the proposed approach requires that residual risk (i.e., the risk remaining after applying risk reduction measures) is continuously monitored and periodically reviewed. Indeed, the hazard identification should be intended as a dynamic process, which periodically revises the knowledge of hazards and the early available warnings [11,12].

### 2.2.1. The Checklist

The first step of applied methodology consists of a checklist designed to recognize criticalities related to the engineering and biotechnological aspects of the process (e.g., pathogen agents and hazardous substances). Its use is aimed at collecting as much information as possible on the bioprocess, in order to provide a first screening of the operating parameters and conditions to monitor, becoming a preparatory base for a more detailed analysis. In particular, the main outcome of the developed checklist application is the creation of a list of fundamental parameters of bioprocess under analysis. The list is a relevant input in the BioHazOp application.

The proposed checklist is related to specific issues aimed at the identification of the role of the different operating parameters. It consists of two sections:

(1)   Process specification section (engineering process and bioprocess: substances hazard classification, biohazard, flammability, explosivity and relevant parameters);
(2)   General section (management outline): operating procedures, plant layout, emergency response/on-going programs and process hazard analysis.

The checklist was based on the European legislation Directive 2000/54/EC [13] and Regulation EC No 1272/2008-CLP Regulation [14], since its application is also aimed at ensuring that the process and site are complying with standards and requirements.

### 2.2.2. BioHazOp

The second layer of the proposed methodology is based on a modified HAZOP procedure, which will be referred to as BioHazOp, because it is related to biotech processes. Dealing with bioprocesses implies that common deviations could induce consequences, which somehow are new with respect to conventional chemical processes, and that the relationship between causes and consequences (bio and not) requires a specific investigation.

In common with HAZOP, the BioHazOp application is based on detailed information of the plant and operations under analysis. Even in this case, an interdisciplinary team is required and the analysis is carried out by a leader, who guides the team throughout all the possible guidewords coupled with the process parameters of the node under analysis to identify the deviations.

In BioHazOp, no new guidewords were necessary, since the approach of conventional HAZOP studies allowed us to perform the analysis in a satisfactory way.

In the present study, the team prepared Relevant Deviations Matrices (RDMs) as a result of the brainstorming sessions. The RDM combines a standard set of guidewords with a general set of process parameters for the node to be analysed. The RDM scheme is reported in Table 1. The matrix utility consists of listing the possible combinations of guidewords and process parameters, which have to be used in the BioHazOp study.

**Table 1.** Relevant Deviations Matrix (RDM) scheme.

| Process Unit | |
|---|---|
| | process parameter |
| guide word 1 | |
| guide word 2 | |
| guide word n | |

Table 2 shows the BioHazOp flowsheet. Once the node and the parameters are selected, the deviations from normal operating condition are initially analyzed from an engineering standpoint, looking for causes, consequences and countermeasures. The biotechnological process aspects are successively taken into account, in the same way, highlighting causes and consequences related to the microorganisms and their behavior (namely "biocauses" and "bioconsequences").

**Table 2.** BioHazOp worksheet scheme.

| Process Unit | | | | | | |
|---|---|---|---|---|---|---|
| | Engineering process | | Biotechnology process | | Existing counter measures | Biohazard | Proposed counter measures |
| process parameter | causes | consequences | causes | consequences | | | |
| deviation 1 | | | | | | | |
| deviation 2 | | | | | | | |
| deviation n | | | | | | | |

Even though the flowsheet is aimed at focusing the discussion on the specific causes and consequences of biological process deviations, it should be highlighted that a biocause can also determine an engineering consequence and vice versa. This methodological approach induces the BioHazOp team to have a deeper insight on the relations between bio and not-bio parameters, raising more questions than the classical HAZOP. In the study, the presence of biohazards as a consequence of each specific deviation has been assessed. This requirement is satisfied by the introduction of a specific column in the BioHazOp worksheet (Table 2).

### 2.3. Piping and Instrumentation Diagram (P&ID)

The piping and instrumentation diagram is an articulate plant drawing, which includes the piping and process equipment with its instrumentation and control systems. In particular, it shows how process equipment is connected and represents flows directions, safety and control systems and instrumentation details by specific symbols. These must be simple and easy to remember, while, at the same time, clearly depicting the equipment function. A P&ID should generally include mechanical equipment (reactors, tanks, and pumps), valves and their identification, piping, vents, instrumentation (level, temperature and pressure transmitters), etc.

In order to draw a P&ID, modern software can be used. It offers wide symbols libraries and is able to check the entire design for avoiding errors during the drafting process. The diagram is the basis for the safety systems development. In the case study, the software M4 P&ID FX has been used to represent the second node (SBR).

### 3. Case-Study: Pilot Plant for the PHA Production

The BioHazOp analysis has been applied to a pilot plant for the PHA production from municipal organic waste. This plant is located at the municipal wastewater treatment plant (WWTP) of Treviso.

The three-stage process consists of:

(1) Acidogenic fermentation of the organic feedstock for volatile fatty acids (VFA) production;
(2) PHA-storing microorganisms' selection from MMC;
(3) PHA accumulation maximization.

The PHA polymerization is naturally performed by bacteria cultivated under unbalanced growth conditions, when an essential nutrient for growth is present in limited amount in the cultivation medium, whereas carbon is in excess. This carbon storage is used by bacteria as an alternate source of fatty acids, metabolized under stress conditions, and is the key mechanism for their survival.

In order to achieve a high PHA storage performance in the accumulation stage in terms of high PHA yields, the efficiency of the culture enrichment stage is crucial to select high-performing PHA-storing microorganisms. The parameters, which can influence the selection performances, are the solid retention time (SRT), the hydraulic retention time (HRT), the cycle length (CL), the organic loading rate (OLR) and the dissolved oxygen (DO). In the case study, different temperatures and feedstock compositions were initially tested, as well as the effect of thermal hydrolysis. The mesophilic fermentation (37 °C) on thermally hydrolysed feedstock (a thermal pre-treatment is applied and consists of the application of a high temperature (72 °C) for 48 h to the feedstock mixture inside the fermentation reactor. After this time, the reactor temperature is decreased and maintained at 37°C for four days) ensured stability in terms of VFA production at high concentration (30 ± 2 $gCOD_{VFA}$/L) and $COD_{VFA}/COD_{SOL}$ ratio (0.86 ± 0.09) [15]. Figure 2 shows the steps of PHA production process.

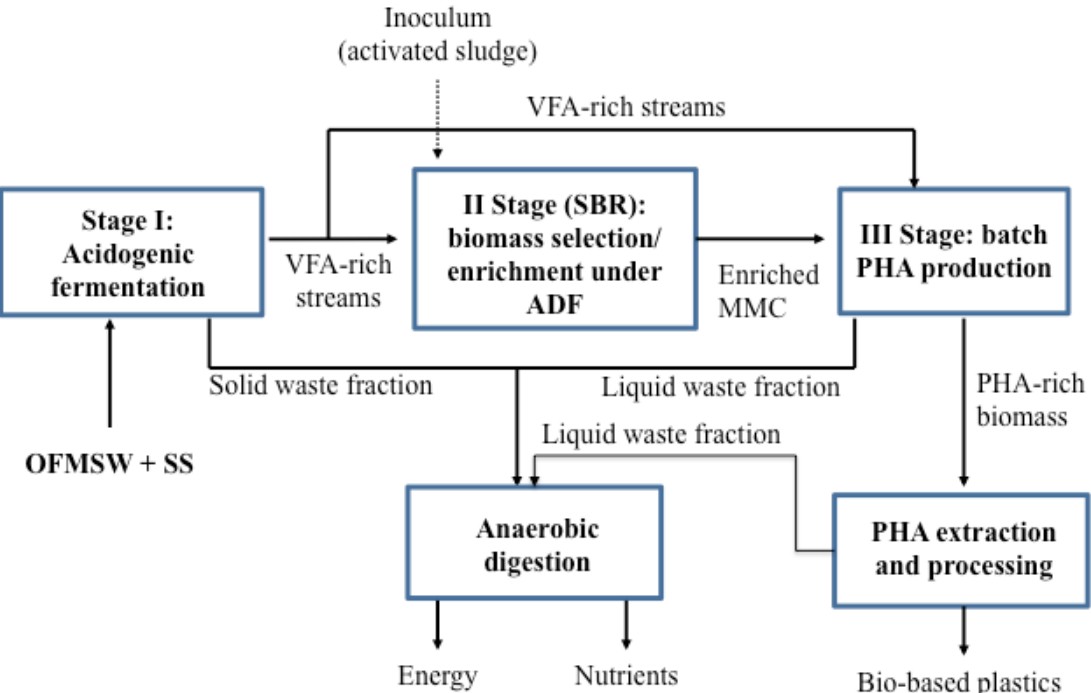

**Figure 2.** PHA production process.

The BioHazOp analysis has been applied to process conditions corresponding to the maximum PHA content in the biomass. This choice has led to the adoption of the following operating parameters (Table 3).

**Table 3.** SBR operating parameters.

| HRT | Window Time (d) | Feedstock | T (°C) | pH |
|---|---|---|---|---|
| 1 | 0.5 | VFA-rich stream | 25–28 | uncontrolled |

The first stage consisted of the acidogenic fermentation of a mixture composed by 30% *v/v* SS and 70% *v/v* OFMSW for VFA production (mixture of the following acids: acetic, propionic, butyric, valeric, isobutyric, isovaleric, caproic, isocaproic and heptanoic). The fermenter was a continuous stirred tank reactor (CSTR). The stream from acidogenic fermentation underwent an additional solid/liquid separation step before being used for feeding the SBR. Indeed, after passing the centrifuge filter equipped with 5–10 μm porosity nylon filter bag, the filtrate was successively filtered by a tubular ceramic membrane with 0.2 μm porosity. The fermented VFA-rich stream was successively fed to the SBR (Figure 3) for PHA biomass selection and to the PHA accumulation reactor. SBR was utilized for culturing the activated sludge under periodic feeding conditions. Each operating cycle was divided into four aerobic phases:

(1) biomass withdrawal;
(2) regeneration;
(3) feeding;
(4) reaction.

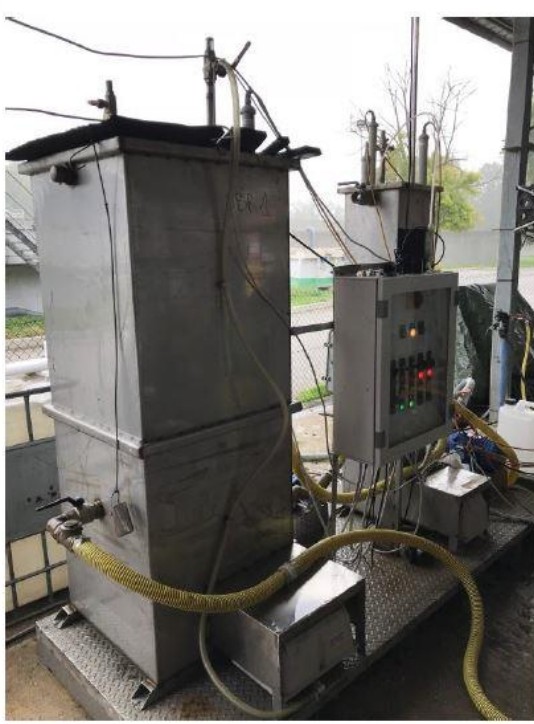

**Figure 3.** Sequencing batch reactor at pilot scale.

The SBR was aerated by linear membrane blowers (Bibus EL-S-250), which allowed an adequate stirring. The SBR temperature and pH were continuously measured, but not actively controlled. The temperature was maintained between 25 °C and 28 °C by an immersion heater. Since no settling phase was programmed, the HRT was equal to the SRT in all runs. The pH was maintained around 8.0 during the whole SBR cycle, having the fermented feedstock a good buffering capacity due to the high alkalinity level (4.4–5.9 g/L $CaCO_3$).

The SBR performance was monitored by measurement of biomass concentration, as volatile suspended solid (sample taken at the end of the cycle), PHA and $COD_{SOL}$ (sample taken at the end of the feast phase and at the end of the cycle).

Both reactors for selection and accumulation worked under a fully aerobic regime and were automatically operated by a programmable logic controller (PLC, MyRio Labview from National Instrument), which also acquired real-time signals from immersion probes. The SBR operated and was monitored under four different runs with an intermediate OLR (4.0 g COD/L d) and a real substrate at the pilot scale.

The microbial community was exposed to alternate feast and famine regimes to enforce the domination of PHA accumulation strains. The external carbon source was provided in abundance during the feast phase, while it became exiguous in famine phase. During the famine phase, the expression of the growth-associated enzymes was reduced and when the organic substances became accessible again, the bacterial strains, which are capable of compiling PHAs as carbon storage, became dominant [16]. This step is known as culture enrichment, which is very crucial, because the dominant microorganisms could influence the PHA yield and composition. Several parameters, such as feedstock composition, organic loading rate (OLR) and the feast and famine ratio (F/F ratio), are extremely significative in the culture enrichment step [17].

The most performing run (Ae3) in terms of storage yield and maximum PHA content was selected. Table 4 shows the operating parameters of Ae3 SBR run.

**Table 4.** Operating parameters applied to Ae3 SBR run.

| Run | HRT (d) | SRT (d) | OLR (g COD/L d) | CL (d) | SRT/CL (d/d) | Feeding Frequency (d$^{-1}$) | Operation Lengt (d) | Load per Cycle (g COD/L) |
|---|---|---|---|---|---|---|---|---|
| Ae3 | 1 | 1 | 4 | 0.5 | 2 | 2 | 45 | 20 |

The PHA content in the biomass (g PHA/g VSS) was determined as the ratio between PHA concentration (g/L) and volatile suspended solid (VSS) concentration (g/L), both at the end of the feast phase. In the selection/enrichment stage, Ae3 run produced a PHA content of $0.59 \pm 0.03$ g PHA/g VSS. For the same run, the overall yield, calculated in terms of storage yield and maximum PHA content achieved, was 110 g PHA/kg VS. In order to verify the full agreement with existing and future regulatory frameworks, it was evaluated which contaminants are of possible concern, when using urban bio-waste as a renewable resource to be transformed into bio-products and bioplastics. Therefore, in PHA samples produced in the pilot-scale process, selected families of contaminants have been searched. They included heavy metals, poly-aromatic hydrocarbons and polychlorinated biphenyls.

The HAZOP analysis was focused on SBR (second node). It has to be noted that on an industrial scale, the reactor is structurally similar to the aeration tank of wastewater treatment plants.

## 4. Results

*Relevant Deviations Matrices and BioHazOp Worksheets*

For the purposes of the BioHazOp analysis, all the information on biological and chemical hazard, as well as on process parameters and on the prevention measures implemented at the Treviso pilot plant, have been collected for the node under analysis.

Table 5 shows the main analysis results, while Table 6 contains information collected with reference to the overall process (whole plant).

A list of relevant deviations for the process under analysis has been drawn up for the node. As a consequence of the brainstorming sessions, the original lists have undergone several revisions. Regarding the bio-parameters, the ones usually taken into account in the study of bioprocesses (enzymatic activity, foam, biochemical oxygen demand, oxidation-reduction potential, conductivity, osmolality and turbidity) have been discussed with respect to hazard and operability, but none has been considered relevant for the case study. Specifically, focusing on the main parameters of the process under study and exploring all the possible combinations of guidewords and parameters to be used in the BioHazOp analysis, a RDM has been created for the node to obtain the set of relevant deviations, which have to be analyzed by the BioHazOp team. The customized RDM, derived from Table 1, is shown in Appendix A (Table A1). Based on the information collected for the whole plant as well as for the node and the RDM, the HAZOP team conducted the BioHazOp analysis. The filled worksheet, derived from Table 2, is reported in Appendix A (Table A2).

Finally, Figure A1 (Appendix A) shows the SBR P&ID, which highlights the process equipment (tanks, reactors, pumps and valves) connections and represents the flows directions. The diagram is the basis for the full-scale plant development. Indeed, its analysis allows us to indicate the instrumentation which is able to measure parameters such as flow, temperature, pressure, etc., and alarm levels linked to main operating parameters and therefore the process can be improved in terms of safety and operability (PHA production).

**Table 5.** Node 2: SBR (collection of useful information for BioHazOp analysis).

The activated sludge from the full-scale plant is used as inoculum for the SBR (aerobic PHA microorganisms selection reactor), and the biomass was selected using an aerobic dynamic feeding strategy (feast–famine regime). Activated sludge is automatically fed to the SBR, a 100 L working-volume reactor.

The dissolved oxygen (DO) concentration is maintained at a maximum of 8.0 mg $O_2$/L with linear membrane blowers, which also allow the complete stirring of the mixed liquor. The DO concentration, oxidation reduction potential (ORP), pH and temperature are constantly monitored in real time by immersion probes and online signals are acquired through a programmable logic controller (PLC). The SBR temperature is regulated by an immersion heater and maintained between 25 °C and 28 °C. SBR operating cycles were automated and controlled by the PLC. The mixture inside the SBR contains:

- Biological agents consisting in the microbial consortium enriched in the SBR growth conditions. NGS-16S and in situ microbial detection methods have been applied for the biomolecular characterization of selected biomass in order to monitor the long-term process robustness and reliability in the SBR. The microbial characterization of PHA-producing bacteria gives inputs for the evaluation of biosafety issues.
- Chemical agents consisting in VFA (acetic acid, propionic acid, butyric acid, valeric acid, isobutyric acid, isovaleric acid, caproic acid, isocaproic acid and heptanoic acid). Such substances pose physical hazards and/or health hazards according to the [14] EC Regulation 1272/2008 (CLP Regulation) classification criteria.

Unwanted reactions may be caused by abnormal process conditions (temperature, pH), abnormal flow rates, system leakage, electric equipment malfunction (e.g., sensors) or mechanical failure (e.g., pump or blower trip).

The process equipment includes drains (sampling points).

The process does not work in or near the flammability range.

Further detailed information has been acquired with respect to the following:

-BIOHAZARD: it is connected with PHA producing microorganisms enriched in the SBR. The microbial consortium, enriched in the SBR growth conditions, includes many microbial genera. The microbial characterization has identified about 200 bacterial genera as reported in Crognale et al. [18], some of which may contain species assigned to the risk group 2 (low pathogenicity), according to the European classification (Annex III of Directive 2000/54/EC [13] as amended by Directives 2010/1833 [19] and 2020/739 [20]).

The exposure routes to the biological agents present in the SBR include:

- Oral intake from splashes, eating, drinking, smoking and any hand-to-mouth contact via contaminated clothing or personal protective equipment;
- Intake through the respiratory system (inhalation) from bioaerosols (e.g., droplets or dust particles);
- Intake through the skin or mucous membranes;
- Penetration into deep tissue by contaminated objects and devices.

-TOXICITY and ECOTOXICITY: the toxic agents in main quantity are $CH_3COOH$ (acetic acid, CAS No. 64-19-7) and $CH_3CH_2CH_2COOH$ (butyric acid, CAS No. 107-92-6) having potential health effects both acute and chronic. The substances entry routes are inhalation, skin contact, and eye contact. Specific toxic effects are associated with each substance. For example, the main toxic effects of acetic acid are:

(1) Acute effects: increasing concentration involves increasing corrosive effects on skin and mucous membranes, and exposure to high concentrations causes severe damage to the eyes and the lungs. The oral intake of high concentrations can cause chemical burns in the digestive tract, metabolic disorders, blood impairment, cardiovascular reactions and renal damages.
(2) Chronic effects: skin changes, chronic inflammation of eyes and respiratory tract and erosive tooth damage. Some substances (valeric acid, hydrogen sulphide) are ecotoxic for the aquatic systems.

Preventive and protective measures include the use of gas detectors and of personal protective equipment for tasks involving the chemicals handling.

-FLAMMABILITY and EXPLOSIVITY: the SBR does not contain flammable gaseous mixtures.

**Table 6.** Overall process (whole plant)—collection of useful information for BioHazOp analysis.

| Operating procedure |
| --- |
| There is no specific written procedure to maintain the on-going integrity of the process equipment. Nevertheless, good operating practices have been applied to plants operations, such as cleaning of coaxial centrifuge and ultrafiltration membranes, and taking samples for laboratory analysis. |
| **Plant layout** |
| There are buffer zones between the plant and the external public (population). The pilot plant operators are potentially exposed to hazards from the wastewater treatment plant within which the PHA production process takes place. The OFMSW transfer to digester is carried out under containment conditions and therefore it has no impact on the environment and operators. The workplace layout includes the location of control rooms, laboratories and offices, drainage areas and sampling points. |
| **Emergency/ongoing program** |
| A number of steps in the production process are managed by programmable logic controller (PLC). There is not any system, which ensures the plant is currently kept and periodically tested. |
| **Management—Process Hazard Analysis (PHA)** |
| No risk analysis techniques other than BioHazOp analysis (FTA, FMEA, What if) have been applied to the plant. |
| - BioHazOp addresses the following; |
| - Hazards of the process; |
| - Process equipment; |
| - Engineering and administrative controls; |
| - Consequences of failure for engineering or administrative controls, including consequences of deviation and steps required to correct or avoid the deviation; |
| - A qualitative evaluation of safety and health effects (of failure) on employees in the workplace. |
| The BioHazOp analysis has been performed by a team, that had the expertise: |
| - In engineering and process operations; |
| - In the BioHazOp evaluation methodology; |
| - In biological and chemical (health and safety issues) risk assessment. |

## 5. Discussion

The BioHazOp analysis has been focused on the most efficient SBR run in terms of maximum PHA content, among the ones conducted in the Treviso pilot plant [8]. The most significant result of the analysis is the identification of technical solutions and management measures, which have to be applied to the plant for ensuring its operability over time and the occupational risks (they are due to the possible contact with biological and chemical agents during the work activities) mitigation. As reported in the last right-hand column of Table A2 for almost every pair of cause-consequences, safeguards, which could prevent, detect, control or mitigate the hazardous situation, have been identified. Finally, if the safeguards have been considered insufficient to solve the problem, additional technical solutions and recommendations have been proposed. In filling the worksheets for the BioHazOp analysis during the brainstorming sessions, some of the selected parameters, guidewords and deviations have been eliminated or modified in accordance with the in-depth analysis. With reference to SBR, the cycle length has not been examined.

A very important outcome of BioHazOp analysis is the attention, which has to be addressed to valve (valves equipped with electric actuators and non-return valves) maintenance in order to ensure the process efficiency. The maintenance and its timing depend on service conditions (temperature, pressure, fluid typology, etc.). The periodic cleaning is recommended and the actuator must not be cleaned by aggressive solvents or highly flammable detergents or those and injurious to health. During and after cleaning, the sealing points on the actuator should be inspected. In case of lubricant loss and accumulated

dirt, the sealing elements have to be repaired. In the case of hydraulic non-return valves provided with dampers, a specific maintenance is required. It consists of:

- A periodic check of their side covers. In case of leakages, the screws must be re-tightened or gaskets must be replaced;
- A hydraulic damper check (lubricant level). In particular, it is recommended to use lubricant, which has a kinematic viscosity included between 30 and 50 mm$^2$/s (ISO grade);
- Check of hydraulic circuit connections (all the components have to be perfectly tightened).

Once a month, it is recommended to check the correct valves operating by their opening and closure. Before carrying out the valves maintenance, the line has to be drained and depressurized and the electric supply must be disconnected. The non-observance of these precautions could cause injuries to workers. In particular, it is extremely important that the operators responsible for the valves installation, operation and maintenance are properly qualified and trained.

With reference to the pilot plant, another analysis outcome consists of the need of monitoring the VFA concentration. A common system to measure VFA concentrations is to apply chromatographic techniques to sample aliquots taken in the mixture. In order to check the fermenter efficiencies and VFA yields, gas chromatography equipped with a capillary column and flame ionization detector has been widely and successfully applied. The difference in nature and complexity of the matrices, where the VFA can be present, resulted in the development and publication of a great number of procedures for VFA quantification and speciation [21]. An innovative analytic approach is presented for determining the total volatile fatty acids concentration in anaerobic digesters [22]. This technique is adequate for the purpose of determining the total VFA and for alarming operators in case of process deterioration and imminent failure. It is simple to execute and may be used by researchers working on anaerobic processes, but it is also appropriate as a routine tool for controlling full-scale anaerobic digesters.

For the purpose of occupational risk management, further investigation should be conducted with respect to operations usually carried out at the plant, such as the feeding and discharge of fluid mixtures from pumps, sampling of the liquid mixture for laboratory analysis, membrane filtration operations and more generally solid–liquid separation steps, membrane washing by chemical agents (sodium hydroxide, sodium hypochlorite, phosphoric acid) and plant shutdown for the pumps and filters maintenance. As mentioned in 2.3, the P & ID is a plant drawing, including the piping and process equipment with its instrumentation and control systems. P&ID plays a strategic role in the maintenance and modification of process, which it describes, and it provides the basis for the development of system control schemes, allowing safety and operational investigations. Therefore, it represents a fundamental milestone in order to ensure the best performance of industrial processes in terms of operability, efficiency and safety. These goals can be achieved by the rigorous choice of instrumentation and alarm levels linked to main operating parameters (reagents flow, temperature, pressure). Indeed, in order to control the PHA production process and avoid parameters deviations from design intention, the instrumentation and alarm levels play a particularly relevant role. Another fundamental aspect in the industrial processes running is the machines redundancy, which is required to avoid operability discontinuities. With reference to SBR, the blowers' redundancy is crucial to achieve an uninterrupted mixing, which is a very important requirement in order to avoid the biomass sedimentation and decreased substrate degradation. Indeed, these two phenomena could decrease the expected PHA production. All hazards not identified in the HAZOP analysis are not considered in terms of risk and this is particularly relevant for biological risks, whose assessment is difficult in the workplace.

In this case study, biohazard is connected with PHA-producing microorganisms enriched in the SBR. They have been characterized at the genus level by culture-independent methods, based on 16S rRNA gene high-throughput sequencing as reported in Crognale et al. [18]. The microbial consortium, enriched in the SBR growth conditions, includes many

microbial genera. The microbial characterization has identified about 200 bacterial genera, some of which may contain species assigned to the risk group 2 (low pathogenicity), according to the European classification (Annex III of Directive 2000/54/EC [13] as amended by Directives 2010/1833 [19] and 2020/739 [20]). Some species of the microbial detected genera, such as Acinetobacter, Pseudomonas, Aeromonas, Comamonas and Brevundimonas, have been previously reported as potential human pathogens and they should be considered opportunistic agents, which do not cause any infections to healthy employees, but they can generate diseases when the body defenses are defective. Their pathogenicity mainly occurs by contact (infection of wounds) or by inhalation. During some process activities in the PHA plant (such as handling of the electromechanical pumps, pipes, compressors, valves, drainage, cleaning equipment, cleaning tasks, dryers and conveyors cleaning and maintenance), the sensitizing and toxic risk related to the exposure to biological agents should be taken into account. Compared with the workers of the wastewater treatment plants, the operators involved in the PHA production process from biowaste may be exposed to bioaerosols, aeroallergens and biological components conveyed such as particulates (i.e., bacterial endotoxins, fungal spores), which can cause a burden of pulmonary diseases [23]. Therefore, the workers' activities should be checked to define the exposure characteristics. The potential occupational exposure could be identified and documented through the biological environmental monitoring plan, but it is not mandatory according to Directive 2000/54/EC [13].

The biological risk assessment in activities, which are not allocated to a safety level, such as laboratory activities, is seriously hampered, since neither universally approved criteria for assessing the exposure to biological agents, nor agreed dose–response estimates and occupational exposure levels (OELs) are available [24]. The occurrence of a variety of PHA accumulating bacteria, which ensure a stable PHA production in the open system, such as the SBR, would require the necessity of assessing the workers exposure to bioaerosol, but the selection of the relevant parameters is not simple, because of its complexity. Furthermore, the microorganisms viability is less important for effects, such as chronic bronchitis, asthma, toxic pneumonitis, hypersensitivity pneumonitis and lung function decline, as these effects may also develop after exposure to non-viable microorganisms [23].

Currently, there are no universally accepted air sampling and analysis methods for quantifying the workers exposure to bioaerosol and there is still a lack of health-related exposure limits based on toxicological or epidemiological studies from environmental health or the workplace [25]. The potential variation of microbial composition, such as in the bioprocesses based on MMC, adds difficulties to assess the occupational biohazard. Moreover, the microorganisms constantly interact with the environment and are able to modify their pattern of gene expression rapidly in response to environmental signals [26,27]. Another challenge for the microbial risk assessment is the emergence and rise of antibiotic resistance observed worldwide. Indeed, an unexpected density of antibiotic resistance genes has been discovered by different metagenomic studies in soil and in water habitats [28]. Regarding the host response, the role, which is played by the biological agents in the development or aggravation of symptoms and diseases, is poorly understood. The human response to exposure to biological agents depends on the organic involved material and individual susceptibility to infections and allergies.

In order to overcome the current knowledge gaps in the biological risk assessment, the potential risk should be managed in a precautionary manner, taking into account, by an expert in microbiology, the biological agents involved in the biotech process and biohazard related to plant areas and workers activity [29].

Since the microorganisms are an inherent part of the bioprocess, the hazard cannot be eliminated, but the workplaces and work processes design and the choice of adequate equipment and working methods allow the control of the occupational biohazard in the PHA production plant. In order to prevent the infectious workers risk, effective personal hygiene measures are sufficient, including the provision of adequate hand washing facilities. The sensitizing and toxic effects can be controlled by minimizing the generation of

bioaerosols and dust in the workplace. Where residual hazards/risks cannot be controlled by collective measures, the employer should provide for appropriate personal protective equipment, such as suitably fitted respiratory devices, when working in areas where bio-aerosol is generated.

In a full-scale plant, the employers should demonstrate that adequate control measures have been developed in accordance with the controls hierarchy [30,31] in order to minimize the workers exposure to biological agents. The types of measures can be classified as:

(1) Technical measures (e.g., aeration of SBR performed by air bubbles under pressure, instead of surface aerators);
(2) Organizational measures (e.g., only qualified employees are allowed to do specified works; signage/warnings and/or administrative controls);
(3) Personal measures, such as personal protective equipment (PPE).

In a full-scale plant, any activities involving the movement of organic matrices, waste and/or sludge should be controlled to avoid the organic dust and/or bioaerosol release in the workplace. With regard to workers exposure control, several preventive measures should be implemented:

- Using air diffusers, such as an aeration system, instead of systems involving the mechanical agitation of fermentation liquid;
- Isolating workers from equipment with high bioaerosol emissions and using glass panes or plastic curtains;
- Forced ventilation system for activities located within indoor places;
- Organizing the work tasks to prevent workers from spending long periods in areas where the bioaerosol concentrations are higher;
- Providing adequate welfare facilities and first-aid equipment;
- Ensuring the use of PPE, such as gloves, goggles, a face shield, water-resistant suit or respiratory protective equipment (RPE), depending on the worker task;
- Making effective arrangements for monitoring the staff health.

The equipment cleaning tasks are identified as high priority activities, because they could generate bioaerosol emissions and potential exposures for workers. The dryers and conveyors cleaning and maintenance require equipment is opened, and therefore it can result in higher personal exposures to bioaerosol [32]. Evidence from epidemiological data shows that work-related exposure to bioaerosol may cause a number of different health problems, including respiratory diseases, skin problems and gastrointestinal symptoms [33]. Thus, it is recommended that for exposure to bioaerosols, RPE is provided with the highest efficiency filters (P3). This rule should be applied within a radius of 50 m from the operational areas, considering that bioaerosol levels typically return to background concentrations within this distance [34]. Risk of infection and some of the symptoms (e.g., flu-like symptoms) may be preventable by introducing relatively simple hygienic measures [35]. The employer must provide the workers with the procedures for sampling, handling and processing the contaminated matrices. The employers should undertake an appropriate health surveillance of their workforce to ensure that health outcomes related to exposure to biological agents are managed and reported. Some pathologies related to biological agents are poorly specific and it is extremely important that workers report any work-related symptoms to the employer and the physician (Article 10 of Directive 2000/54/EC). Among the prevention measures, a relevant role is played by information and training of workers. The employer must take all necessary measures to ensure that employees are trained and properly informed on potential risks to health and precautions to be taken to prevent the exposure.

## 6. Conclusions

BioHazOp is a strategic tool for improving the industrial processes safety and operability. This analysis is able to find the adequate countermeasures to avoid the parameters deviations from design intention. The deviations could cause a process efficiency decrease

and hazardous scenarios (toxic gasses releases, exposure to biological agents, etc.) for workers' health and safety. The analysis results clearly show that the proposed methodology allows a comprehensive exploration of conventional hazards and biological hazards. Indeed, the conventional hazards are generally more familiar to the team members and may be easily overlooked by process experts usually dealing with lab scale bioprocesses. On the other side, the biological risk assessment in the biorefinery process from organic waste is a complex task, even considering that this industrial sector is still in its infancy and limited public domain information is available from the workers health surveillance.

The biological risk assessment in activities, which are not addressed to a safety level, such as lab activities, is difficult, since neither universally approved criteria for assessing the exposure to biological agents, nor agreed dose–response estimates and occupational exposure levels are available. In order to overcome the current knowledge gaps in the biological risk assessment, the potential risk in the biotech plant should be managed through a preventive approach. The proposed approach for the biological risk control of the PHA production process is that certain areas or process activities can be categorized, using fairly simple descriptive expressions of risk and a corresponding set of control measures, which depend on the perceived risk associated with the area or the activity. For exposure to biological agents with low pathogenicity (risk group 2 according to the Annex III of Directive 2000/54/EC), as for the working activities in the PHA production process, the risk estimation can be mainly carried out in epidemiologic terms, i.e., observing, because of exposure (even only presumed), the incidence of workers health outcomes through the health surveillance. The assessment of diseases, which may be contracted, should be based on all available information for the examined workplace or related sectors (i.e., wastewater treatment). Epidemiological available data in wastewater treatment plants show that, while the risk of contracting an infection is generally low, respiratory, gastrointestinal and flu-like symptoms are more common in wastewater treatment workers [36–38]. Therefore, the design of workplaces and work processes, the choice of adequate equipment, the identification of working procedures and best practices allow the control of occupational biohazard in the bioprocess production plants.

In particular, BioHazOp is a tool, which is able to take into account the biotechnological process aspects, highlighting causes and outcomes related to the microbial consortia and their behaviour. This approach can generate a significant improvement of biotechnological processes safety and operability, because the usual HAZOP analysis does not investigate the possible interactions between engineering and biotechnological aspects.

**Author Contributions:** Methodology, R.L., E.I. and B.P.; formal analysis, R.L., E.I., B.P., L.N.T., F.V., M.G. and M.M. All authors have read and agreed to the published version of the manuscript.

**Funding:** This research received no external funding.

**Institutional Review Board Statement:** Not applicable.

**Informed Consent Statement:** Not applicable.

**Data Availability Statement:** In this section, please provide details regarding where data supporting reported results can be found, including links to publicly archived datasets analyzed or generated during the study.

**Acknowledgments:** This work was supported by the REsources from URban BIowaSte"—RES URBIS (GA 7303499) project in the European Horizon 2020 (Call CIRC-05-2016) program.

**Conflicts of Interest:** The authors declare no conflict of interest.

## Appendix A

**Table A1.** Relevant Deviations Matrix (RDM) listing the possible combinations of guidewords and process parameters to be used in BioHazop study (node 2).

| | | | | | | | | | | |
|---|---|---|---|---|---|---|---|---|---|---|
| **RELEVANT DEVIATIONS MATRIX** | | | | | | | | | | |
| **Parameters** | | | | | | | | | | |
| **SBR** | | | | | | | | | | |
| **Guide words** | Flow (VFA-rich stream) | Temperature | Mixing | CL (cycle length) | HRT | HRT/CL | DO (dissolved oxygen) | Organic Loading Rate (OLR) | Water flow | Withdrawal |
| more | X | | | | | | | | X | X |
| no/less | X | | | | | | X | | X | X |
| reverse | X | | | | | | | | X | |
| lower | | X | | | | X | | X | | |
| higher | | X | | | | X | | X | | |
| no | | | X | | | | | | | |
| longer | | | | X | X | | | | | |
| shorter | | | | X | X | | | | | |

**Table A2.** Worksheet obtained by BioHazOp analysis of node 2 (SBR: Sequential Batch Reactor).

| Node 2: SBR | | | | | | | |
|---|---|---|---|---|---|---|---|
| Deviation | Engineering process | | Biotechnology process | | Existing counter measures | BioHazard | Proposed counter measures |
| | causes | consequences | causes | consequences | | | |
| Working volume | | | | | | | |
| MORE | -failure of IE2 -failure of E7 -failure of IE3 and E6 -failure of IE4 and E11 -failure of PLC control system | -increased VFA concentration -HRT increase due to increased SBR working Volume [1] -uncontrolled release of substrate from the tank | | -total or partial inhibition of microorganisms that may increase the organic load [2] -decrease of the rate of PHA synthesis -selection of a slower and less efficient microbial community in terms of PHA production | -visual inspection -periodic maintenance of IE2 and E7 -IE2 and E7 setting check -maintenance of the level gauge installed on the tank [3] | | -install a flow meter on the E7 outlet section |
| NO/LESS | -failure of IE2 -failure of E7 -failure of IE3 and E6 -failure of IE4 and E11 -failure of PLC control system | -decreased VFA concentration -decrease or absence of substrate in the reactor | | -lowering of the organic load -microbial cells decay [4] | -visual inspection -periodic maintenance of IE2 and E7 -IE2 and E7 setting check | | |
| REVERSE | -failure of IE2 -failure of E7 -failure of non-return valve on the E7 outlet section | -absence of substrate in the reactor [5] | | | -visual inspection -periodic maintenance of IE2 and E7 -IE2 and E7 setting check -periodic maintenance of non-return valves | | |
| Temperature (immersion heater) | | | | | | | |
| LOWER (<25°C) | -heater failure | | | -slowing down of biomass kinetics -increased substrate degradation times -increased feast phase with selection damage -decreased PHA yield | | | |

**Table A2.** *Cont.*

| Node 2: SBR | | | | | | | |
| --- | --- | --- | --- | --- | --- | --- | --- |
| Deviation | Engineering process | | Biotechnology process | | Existing counter measures | BioHazard | Proposed counter measures |
| | causes | consequences | causes | consequences | | | |
| Working volume | | | | | | | |
| HIGHER (>28°C) | -heater failure | | | -uncontrolled and different composition of the mixture due to the possible growth of other species of microorganisms [6] <br> -decrease in PHA yield for T>30°C | | | |
| Mixing | | | | | | | |
| NO | -blower failure | -biomass sedimentation <br> -little or no substrate degradation <br> -emptying of the biomass from the reactor by the washout (pump E11) | | | -periodic cleaning of the plates | | -blower redundancy ( installation of a second blower) <br> -electric blower motor provided with preferential power supply (generator or uninterruptible power supply) |
| HRT (1–2 days) [7] | | | | | | | |
| LONGER (>2 days) | -failure of PLD software <br> -failure of IE2 <br> -failure of E7 | | | -ineffective selection of micro-organisms | -visual inspection <br> -periodic maintenance of IE2 and E7 <br> -IE2 and E7 setting check | | -install flow transmitters |
| SHORTER (<1 days) | -failure of PLD software <br> -failure of IE2 <br> -failure of E7 | | | -inactivation of micro-organisms (washout) | -visual inspection <br> -periodic maintenance of IE2 and E7 <br> -IE2 and E7 setting check | | -install flow transmitters |
| HRT/CL [8] | | | | | | | |
| LOWER | -failure of PLD software <br> -failure of IE2 <br> -failure of E7 | -See SHORTER HRT | | | -visual inspection <br> -periodic maintenance of IE2 and E7 <br> -IE2 and E7 setting check | | -install flow transmitters |
| HIGHER | -failure of PLD software <br> -failure of IE2 <br> -failure of E7 | -See LONGER HRT | | | -visual inspection <br> -periodic maintenance of IE2 and E7 <br> -IE2 and E7 setting check | | -install flow transmitters |
| DO (dissolved oxygen) [9] | | | | | | | |
| NO/LESS | -blower failure <br> -plates clogging | | | -biomass decay | | | -blower redundancy (installation of a second blower) <br> -electric blower motor provided with preferential power supply (generator or uninterruptible power supply) <br> -oxygen sensor redundancy |
| Organic Loading Rate (OLR) [10] | | | | | | | |

**Table A2.** *Cont.*

| Node 2: SBR | | | | | | | |
|---|---|---|---|---|---|---|---|
| Deviation | Engineering process | | Biotechnology process | | Existing counter measures | BioHazard | Proposed counter measures |
| | causes | consequences | causes | consequences | | | |
| Working volume | | | | | | | |
| LOWER (<4) | -failure of IE2 -failure of E7 -failure of PLC software | -feeding of less substrate -progressive lowering of working volume -HRT decrease | | -biomass decay -washout | -visual inspection -periodic maintenance of IE2 and E7 -IE2 and E7 setting check -periodic monitoring of the VFA concentration in the fermented stream -addition of water to the SBR by E6 and IE3 | | -online analysis of the VFA concentration [11] in the storage tank and automatic intervention of feeding system (IE2 and E7) and water addition system (IE3 and E6) to ensure a constant organic load |
| HIGHER (>4) | -failure of IE2 -failure of E7 -failure of PLC software | -substrate feeding increase -excessive VFA concentration -excessive increase of working volume and HRT raising | | -selection inhibition -excessive duration of the feast phase | -visual inspection -periodic maintenance of IE2 and E7 -IE2 and E7 setting check -periodic monitoring of the VFA concentration in the fermented stream -addition of water to the SBR by E6 and IE3 | | -online analysis of the VFA concentration[12] in the storage tank and automatic intervention of the feeding system (IE2 and E7) and water addition system (IE3 and E6) to ensure a constant organic load |
| Water flow | | | | | | | |
| MORE | -failure of IE3 -failure of E6 -failure of PLC software | -biomass dilution -excessive increase of working volume and HRT raising | | | -visual inspection -periodic maintenance of IE3 and E6 -IE3 and E6 setting check | | |
| NO/LESS | -failure of IE3 -failure of E6 -failure of PLC software | -biomass concentration -progressive lowering of working volume and HRT decrease | | | -visual inspection -periodic maintenance of IE3 and E6 -IE3 and E6 setting check | | |
| REVERSE | -failure of IE3 -failure of E6 | | | | -visual inspection-periodic maintenance of IE3 and E6-IE3 and E6 setting check | | Install a non-return valve on E6 outlet section |
| Withdrawal | | | | | | | |
| MORE | -failure of IE4 -failure of E11 -failure of PLC software | -variation in the SBR working volume -see Flow (VFA-rich stream) | | | | | |
| NO/LESS | -failure of IE4 -failure of E11 -failure of PLC software | -variation in the SBR working volume -see Flow (VFA-rich stream) | | | | | |

[1] IE4 and E11 effectively run; [2] The final effect may be the perturbation of the steady state; [3] The level gauge can be clogged by the biomass accumulation; [4] Biomass decay is caused by the absence of substrate for microorganisms; [5] The likelihood of reverse flow is remote due to the presence of several non-return valves on the outlet section of E7; [6] Some studies refer to processes carried out at T up to 30 °C, which are suitable for isolating different microbial communities. However, there are no references to studies at T > 30 °C at which, under aerobic conditions, the biomass produces exopolysaccharides, which make the solutions viscous and can inactivate the cells. [7] In the process carried out in Treviso, HRT = SRT has been set. The HRT value must be as low as possible (1–2 days) in order to select a microbial culture, which is able to produce PHA on a kinetic basis; [8] The optimum ratio for maximizing PHA yield is 2; [9] The maximum dissolved oxygen value can be 8.5 mg/L (saturation condition). Higher concentrations are not toxic to micro-organisms; [10] The best range for the process is included between 3 and 6. The value is chosen according to the mass of substrate, the concentration of VFA and the volume of the reactor. [11,12] See *Results and discussion*.

Figure A1 shows SBR P&ID. In particular, it highlights how process equipment (tanks, reactors, pumps, valves as listed in the left side) are connected and represents the flows directions. The diagram could be the basis of a full-scale plant development.

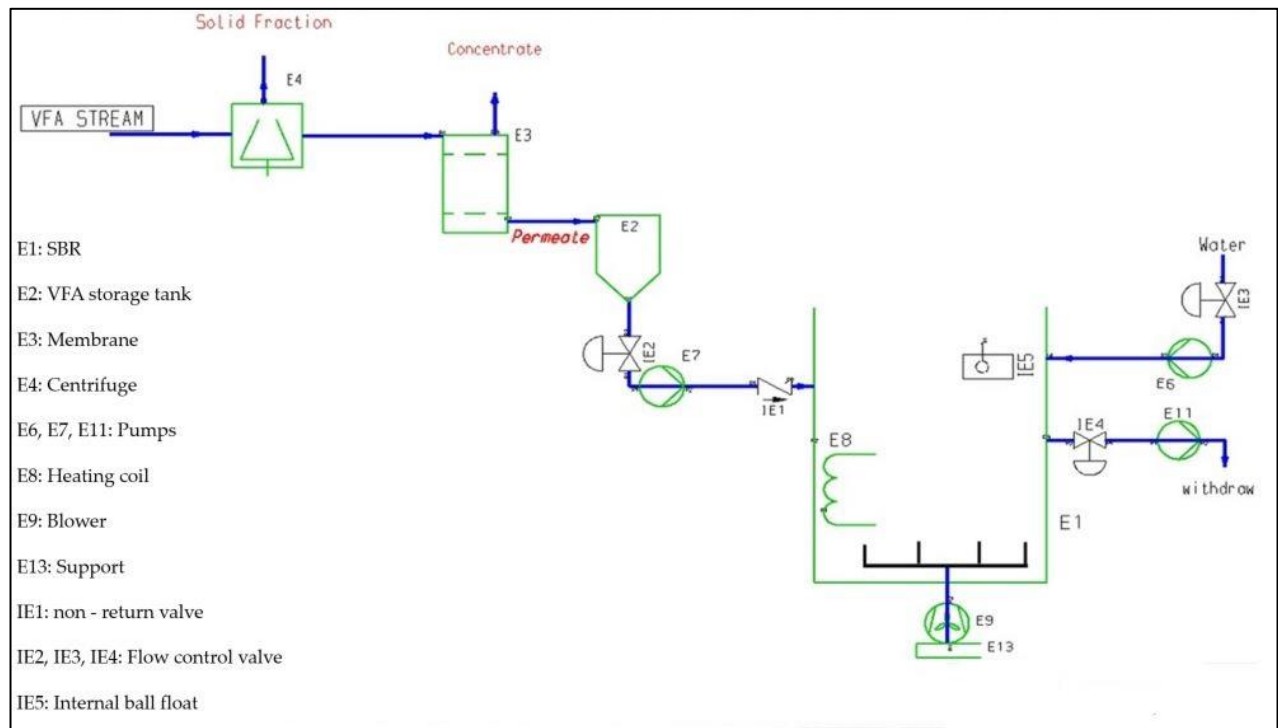

**Figure A1.** SBR P&ID.

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
