# Peer review of "Hazop Analysis of a Bioprocess for Polyhydroxyalkanoate (PHA) Production from Organic Waste: Part B"

_fermentation, doi:10.3390/fermentation9020154_

Round 1

Reviewer 1 Report

This is an interesting article that has been well-researched and is of interest for those working on PHA fermentation technology. The references to legal documentation is good and informative for researchers. 

Some sections could be more concise and to the point and it would also benefit from a native speaker, checking for some of the minor errors throughout. Sometimes the format is inconsistent, using American English and sometimes not, also when figures are referred to, normally capitals are used, sometimes they are not.  

The use of "etc" appears too often, in these cases, 3 or 4 examples are more than enough, so the sentence could end with "X,Y or Z". 

Conclusion, first line has minor errors. Most points are clearly expressed, the authors could mention some of the economic impacts in greater detail and offer some further research/outlook for this particular area of investigation. 

Some of the minor errors:

Line 119 - "this" study

Line 279+ no punctuation. 

Line 340 - and may be used..

Line 471 - "are connected" 

Line 472 - "could be the basis of a full-scale plant..."

Figure A1 could be larger, it would be clearer to understand. 

Author Response

Point 1: This is an interesting article that has been well-researched and is of interest for those working on PHA fermentation technology. The references to legal documentation is good and informative for researchers. Some sections could be more concise and to the point and it would also benefit from a native speaker, checking for some of the minor errors throughout. Sometimes the format is inconsistent, using American English and sometimes not, also when figures are referred to, normally capitals are used, sometimes they are not.  

Response 1: Language has been improved. All figures have been controlled and reported in accordance with the Template.

Point 2: The use of "etc" appears too often, in these cases, 3 or 4 examples are more than enough, so the sentence could end with "X,Y or Z". 

Response 2: as suggested by the reviewer, the use of “etc” has been greatly decreased.

Point 3: Conclusion, first line has minor errors. Most points are clearly expressed, the authors could mention some of the economic impacts in greater detail and offer some further research/outlook for this particular area of investigation. 

Some of the minor errors:

Line 119 - "this" study

Line 279+ no punctuation. 

Line 340 - and may be used..

Line 471 - "are connected" 

 Line 472 - "could be the basis of a full-scale plant..."

Figure A1 could be larger, it would be clearer to understand. 

Response 3: In the paragraph “Conclusions”, the first line has been improved. 

Line 119 - "this" study → the modification has been inserted.

Line 279+ no punctuation → the modification has been inserted. 

Line 340 - and may be used → the modification has been inserted.

Line 471 - "are connected" → the modification has been inserted.

 Line 472 - "could be the basis of a full-scale plant..." → the modification has been inserted.

Figure A1 could be larger, it would be clearer to understand → as suggested by the reviewer, the Figure A1 size has been increased and Lines 504 and 505 have been inserted to explain the figure.

Reviewer 2 Report

 The manuscript entitled Hazop analysis of a bioprocess for poly-hydroxy-alkanoate (PHA) production from organic waste: part Bpresents a Hazard and operability (HAZOP) analysis of a sequencing batch reactor for the production of PHA. The manuscript is written in detail methodology for HAZOP analysis. However, there is some suggestion for improvement before resubmission. The following are main points to be improved:

Comments and Suggestions for Authors

1.     Highlights and an Abbreviation list must be provided. Abbreviations are used in the abstract.

2.     Inappropriate manuscript format, some part is written in normal format, and some use italic format, be consistent.

3.     Some paragraphs are very small, and need to be re-arranged.

4.     Tables in Annex are very long and need to be broken. Table may be formatted in landscape format.

5.     Table 5 and Table 6 necessary to be written in tabulated format?

6.     Basic results for PHA production are missing.

7.     Materials and methods heading is not true only methods are explained.

8.     Section 2.3 explains the piping and instrumentation diagram (P& ID), but the diagrams are missing.

9.     Please tabulate the feedstock parameters and product parameters. Along with please tabulate basic tolerance of hazardous materials in product.

10. Please use cross-reference of figures and tables.

11. Results and discussion are general and need to be improved

Author Response

Point 1:  The manuscript entitled “Hazop analysis of a bioprocess for poly-hydroxy-alkanoate (PHA) production from organic waste: part B” presents a Hazard and operability (HAZOP) analysis of a sequencing batch reactor for the production of PHA. The manuscript is written in detail methodology for HAZOP analysis. However, there is some suggestion for improvement before resubmission. The following are main points to be improved:

 Comments and Suggestions for Authors

  1. Highlights and an Abbreviation list must be provided. Abbreviations are used in the abstract.

Response 1: In accordance with the other manuscript (Part A), submitted to the same special issue and already published, the abbreviations list has not been inserted. During the paper submission, highlights were provided. Abbreviations are also inserted in the Introduction:

  • Line 38 (PHA);
  • Line 49 (MMC);
  • Line 63 (HAZOP);
  • Line 73 (WWTP);
  • Line 80 (P&ID).

Point 2: Inappropriate manuscript format, some part is written in normal format, and some use italic format, be consistent.

Response 2: The manuscript format has been controlled and all italic words have been modified.

Point 3: Some paragraphs are very small, and need to be re-arranged.

Response 3: The paragraph 4.2 has been deleted and its lines (Lines from 312 to 316) have been inserted in the paragraph 4.1

Point 4: Tables in Annex are very long and need to be broken. Table may be formatted in landscape format.

Response 4: Tables A1 and A2 cannot be broken, because the first describes all combinations of guidewords and process parameters to be used in BioHazop study and the second reports all results of HAZOP analysis. If the tables were broken, their significance would be less clear.

Point 5:  Table 5 and Table 6 necessary to be written in tabulated format?

Response 5: Tables 5 and 6 summarize the information gathered through the compilation of some checklists tailored from those reported in the bibliographic reference no. 6. The tabular form responds to the purpose of schematically distinguishing the different types of collected information

Point 6:  Basic results for PHA production are missing.

Response 6: Lines from 273 to 276 have been added. They report the yield of the most efficient run Ae3 (bibliographic reference 8)

 Point 7:  Materials and methods heading is not true only methods are explained.

Response 7: Materials are explained. Indeed, the relevant deviations matrix (Table 1) and BioHazOp flowsheet (Table 2) are fundamental tools for the application of process analysis tecnique and therefore they are real Materials.   

Point 8:  Section 2.3 explains the piping and instrumentation diagram (P& ID), but the diagrams are missing.

Response 8: The second node (SBR) diagram is reported in Figure A1 (Appendix A).

 Point 9:  Please tabulate the feedstock parameters and product parameters. Along with please tabulate basic tolerance of hazardous materials in product.

Response 9: feedstock parameters are reported at line 237. The VFA mixture composition has been added (lines 238 and 239). In order to give information about hazardous materials in the product, lines from 276 to 280 have been added. 

Point 10:  Please use cross-reference of figures and tables.

Response 10: Tables cross-references are inserted in the manuscript. Indeed, Tables A1 and A2 are derived from Tables 1 and 2 (lines from 309 to 311). 

Point 11:  Results and discussion are general and need to be improved

Response 11: In order to improve the discussion, lines from 370 to 387 and from 401 to 406 have been added. Lines from 463 to 467 and from 474 to 476 have enriched the “Conclusions”. The other statements in the discussion and conclusions are necessarily general, because they refer to a full- scale plant, while the case study was focused on a pilot plant. Recommendations for the legal obligations envisaged for a full-scale plant are successively given.

Point 1:  The manuscript entitled “Hazop analysis of a bioprocess for poly-hydroxy-alkanoate (PHA) production from organic waste: part B” presents a Hazard and operability (HAZOP) analysis of a sequencing batch reactor for the production of PHA. The manuscript is written in detail methodology for HAZOP analysis. However, there is some suggestion for improvement before resubmission. The following are main points to be improved:

 Comments and Suggestions for Authors

  1. Highlights and an Abbreviation list must be provided. Abbreviations are used in the abstract.

Response 1: In accordance with the other manuscript (Part A), submitted to the same special issue and already published, the abbreviations list has not been inserted. During the paper submission, highlights were provided. Abbreviations are also inserted in the Introduction:

  • Line 38 (PHA);
  • Line 49 (MMC);
  • Line 63 (HAZOP);
  • Line 73 (WWTP);
  • Line 80 (P&ID).

Point 2: Inappropriate manuscript format, some part is written in normal format, and some use italic format, be consistent.

Response 2: The manuscript format has been controlled and all italic words have been modified.

Point 3: Some paragraphs are very small, and need to be re-arranged.

Response 3: The paragraph 4.2 has been deleted and its lines (Lines from 312 to 316) have been inserted in the paragraph 4.1

Point 4: Tables in Annex are very long and need to be broken. Table may be formatted in landscape format.

Response 4: Tables A1 and A2 cannot be broken, because the first describes all combinations of guidewords and process parameters to be used in BioHazop study and the second reports all results of HAZOP analysis. If the tables were broken, their significance would be less clear.

Point 5:  Table 5 and Table 6 necessary to be written in tabulated format?

Response 5: Tables 5 and 6 summarize the information gathered through the compilation of some checklists tailored from those reported in the bibliographic reference no. 6. The tabular form responds to the purpose of schematically distinguishing the different types of collected information

Point 6:  Basic results for PHA production are missing.

Response 6: Lines from 273 to 276 have been added. They report the yield of the most efficient run Ae3 (bibliographic reference 8)

 Point 7:  Materials and methods heading is not true only methods are explained.

Response 7: Materials are explained. Indeed, the relevant deviations matrix (Table 1) and BioHazOp flowsheet (Table 2) are fundamental tools for the application of process analysis tecnique and therefore they are real Materials.   

Point 8:  Section 2.3 explains the piping and instrumentation diagram (P& ID), but the diagrams are missing.

Response 8: The second node (SBR) diagram is reported in Figure A1 (Appendix A).

 Point 9:  Please tabulate the feedstock parameters and product parameters. Along with please tabulate basic tolerance of hazardous materials in product.

Response 9: feedstock parameters are reported at line 237. The VFA mixture composition has been added (lines 238 and 239). In order to give information about hazardous materials in the product, lines from 276 to 280 have been added. 

Point 10:  Please use cross-reference of figures and tables.

Response 10: Tables cross-references are inserted in the manuscript. Indeed, Tables A1 and A2 are derived from Tables 1 and 2 (lines from 309 to 311). 

Point 11:  Results and discussion are general and need to be improved

Response 11: In order to improve the discussion, lines from 370 to 387 and from 401 to 406 have been added. Lines from 463 to 467 and from 474 to 476 have enriched the “Conclusions”. The other statements in the discussion and conclusions are necessarily general, because they refer to a full- scale plant, while the case study was focused on a pilot plant. Recommendations for the legal obligations envisaged for a full-scale plant are successively given.

Round 2

Reviewer 1 Report

This paper has improved since I reviewed it previously. 

line [45] statistics for 2021 have been quoted, more up to date (2022) data should be available now.  

In terms of grammatical errors and minor corrections, these seem to have been attended to by the authors. 

Author Response

Point 1: This paper has improved since I reviewed it previously. 

line [45] statistics for 2021 have been quoted, more up to date (2022) data should be available now.  

In terms of grammatical errors and minor corrections, these seem to have been attended to by the authors. 

Response 1: With reference to global PHA market, new data have been inserted (lines from 44 to 45).

Reviewer 2 Report

Accepted

Author Response

Dear Reviewer 2,

Thank you for the useful suggestions. Best regards. Lauri R.